# Convergence and efficiency of global bases using Proper Orthogonal Decomposition for capturing wind turbine wake aerodynamics

Juan Felipe Céspedes Moreno[1], Juan Pablo Murcia León[1], Søren Juhl Andersen[2]

[1]Department of Wind and Energy Systems, Technical University of Denmark, Frederiksborgvej 399, Roskilde, 4000, Denmark
[2]Department of Wind and Energy Systems, Technical University of Denmark, Anker Engelunds Vej 1, Kgs. Lyngby, 2800, Denmark

*Correspondence to*: Juan Felipe Céspedes (jfcm@dtu.dk)

**Abstract.** Wind turbine wakes affect power production and loads, but are highly turbulent and therefore complex to model. Proper Orthogonal Decomposition (POD) has often been applied for reduced order models (ROMs), as POD yields an orthogonal basis optimal in terms of capturing the turbulent kinetic energy content. POD is typically used to understand flow physics and reconstruct a specific flow case. However, Andersen and Murcia Leon (2022) proposed a ROM for predicting wind turbine wake aerodynamics by applying POD on multiple flow cases with different governing parameters to derive a global basis intended to represent all flows within the parameter space. This article evaluates the convergence and efficiency of global POD bases covering multiple cases of wind turbine wake aerodynamics in large wind farms. The analysis shows that the global POD bases have better performance across the parameter space than the optimal POD basis computed from a single dataset. The error associated with using a global basis across the parameter space of reconstructions decreases and converges as the dataset is expanded with more flow cases, and there is a low sensitivity as to which datasets to include. It is also shown how this error is an order of magnitude smaller than the truncation error for 100 modes. Finally, the global basis has the advantage of providing consistent physical interpretability of the highly turbulent flow within wind farms.

**Keywords:** Proper Orthogonal Decomposition, Global Basis, Reduced Order Modelling, Turbulence, Wind Turbine Wake Dynamics.

## 1 Introduction

The proper orthogonal decomposition (POD) is a classic data-driven method for decomposing fluctuations of turbulent flows into orthogonal modes, which provide an optimal linear decomposition in terms of the variance [Lumley (1967); Berkooz et al. (1993)]. POD has been applied on a vast range of flow scenarios, and the POD modes are typically used for one of two main applications. One, the modes can provide a physical interpretation of dominant coherent structures in complex turbulent flow, e.g. [Sirovich (1987), George (1988), Neumann and Wengle (2004), Meyer et al. (2007)]. Two, a truncated set of the modes can be used to construct reduced order models (ROMs), e.g. [Smith et al. (2005), Noack et al. (2011), Semaan et al. (2016), Taira et al. (2017)].

However, an optimal reconstruction in terms of variance might not always be the most desirable basis for creating ROMs. Alternative bases can for instance be derived by changing the norm to optimize other quantities instead of variance e.g. enthalpy, enstrophy, and dissipation [Colonius et al. (2002), Lee and Dowell (2020), Olesen et al. (2023)]. Emphasis can also be on the spectral content by performing POD in the frequency domain using Spectral POD [Sieber et al. (2016)] or the related Dynamic Mode Decomposition [Schmid (2010)], which does not provide orthogonal bases. Furthermore, nonlinear bases can

be formed using autoencoders, which constitute a nonlinear generalization of POD through an artificial neural network (ANN) [Hinton and Salakhutdinov (2006), Vinuesa and Brunton (2022)]. Autoencoders are specifically designed to reduce the number of degrees of freedom required to describe a data set but might lack physical interpretability.

      Irrespective of the decomposition method, the resulting bases are typically applied to data from a single flow case, which corresponds to a single point in parameter space. A single flow case would in the present content correspond to the inflow to

a particular wind turbine in a wind farm operating at a single $C_T$ value [Andersen et al. (2014), Debnath et al. (2017), Bastine et al. (2018), Hamilton et al. (2018)]. However, efforts have been made to transition between different bases to cover different flow cases in parametric studies [Christensen. et al. (1999), Stankiewicz et al. (2017), Xiao et al. (2017)]. Conversely, recent developments [Andersen and Murcia Leon (2022), Fu et al. (2023), Nony et al. (2022), Buoso et al. (2022)] employ a single global basis constructed by applying POD on a combination of multiple flow cases. The global basis maintains the benefits of

POD, namely orthogonality and physical interpretability [VerHulst and Meneveau (2014), Andersen et al. (2017), De Cillis et al (2021)]. Using a global basis for constructing generic ROMs enables consistent physical analysis across different flow conditions using the same basis, and therefore holds the potential for constructing more robust POD models [Bergmann et al. (2009)] including diverse forms of interpolation across parameter space to predict unseen flow cases.

      Previously, Andersen and Murcia Leon (2022) qualitatively compared the resulting global POD modes to local POD modes

derived from individual flow cases, but the efficiency of these bases was not compared. This article quantifies the efficiency of the global POD modes in reconstructing wind turbine wake aerodynamics compared to a local basis for a single flow case. Furthermore, a global POD basis is expected to converge as more flow cases are added [Haasdonk (2013), Hesthaven et al. (2016)], but the selection of which flow cases to include to ensure fast convergence is uncertain. Here, the convergence of the global basis is investigated in accordance with previous studies [Haasdonk et al. (2011), Hesthaven et al. (2016), Quarteroni

et al. (2016)]. The analysis uses a database of Large Eddy Simulations (LES) of wind turbine wake dynamics, which are particularly challenging as they are highly turbulent and include the vast range of turbulent scales in the atmosphere. Therefore, this work contributes by explicitly showcasing the advantages and characteristics of global bases in ROMs applied in a practical, yet complex scenario.

# 2 Methodology

## 2.1 Flow Solver and Turbine Modelling

The LES database is the same as used for creating the predictive and stochastic reduced order model of wind turbine wakes [Andersen and Murcia Leon (2022)], where the simulations were generated using the incompressible finite volume flow solver EllipSys3D [Michelsen (1992), Michelsen (1994), Sørensen (1995)]. A third-order QUICK scheme is used for the convective terms, and a second-order implicit method is used for time stepping. The pressure correction equation is solved with an improved version of the SIMPLEC algorithm [Shen et al. (2003)] and pressure decoupling is avoided using the Rhie-Chow interpolation technique. LES applies a spatial filter on the Navier–Stokes equations, where the smaller scales are modeled through a sub-grid scale (SGS) model to achieve turbulence closure. The Deardorff SGS model is used [Deardorff (1980)]. The turbines are modeled using the actuator disc (AD) method, which imposes body forces in the flow equations [Mikkelsen (2004)]. Initially, the velocities are passed from EllipSys3D to Flex5 [Øye (1996)], which computes the forces and deflections through a full aero-servo-elastic computation, and transfers these back to EllipSys3D [Sørensen et al. (2015), Hodgson et al. (2021)]. The turbines modeling does not include the effects of the nacelle or tower, but this only has a minor influence on the wake-generating thrust [Zhale and Sørensen (2008)].

## 2.2 Simulation Setup

The wind farm is simulated with 14 turbines aligned as shown in Fig.1. The computational domain is $192R \times 20R \times 20R$ in the streamwise, lateral, and vertical directions respectively. The grid is structured and has $3392 \times 192 \times 128 \approx 83 \times 10^6$ grid cells. The grid is equidistant from the inlet to the turbines and in the vicinity of the turbines, where it expands $\pm 4R$ on each side of the turbine center, as well as $4R$ vertically. This equidistant region has a resolution of approximately 20 cells per blade radius, which is highly resolved for AD simulations [Hodgson et al. (2023)]. The grid is stretched towards the lateral, top, and outlet boundaries.

The turbines are separated by 12R in the streamwise direction, and 20R in the lateral direction. Cyclic boundary conditions are imposed on the lateral boundaries to mimic an infinitely wide wind farm. The modeled turbine is the NM80 turbine, which has a radius of R= 40.04m, hub-height of $z_0$= 80m, and rescaled rated wind speed of $U_{rated}$= 14m/s with a corresponding rated power of $P_{rated}$= 2.75MW [Aagaard Madsen et al. (2010)].

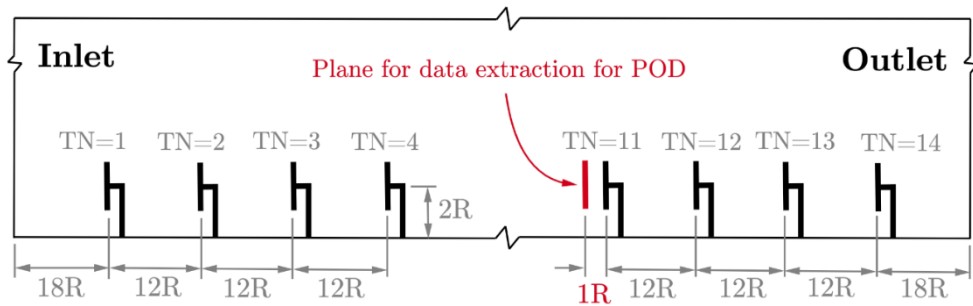

**Figure 1: Simulation layout.**

The neutral atmospheric boundary layer (ABL) inflow to the farm is modeled in a prior precursor simulation [Andersen and Murcia Leon (2022)]. The initial precursor simulation has a roughness of $z_0 = 0.05$m and a friction velocity of $u_* = 0.4545$ m/s, which resulted in an average shear exponent of $\alpha = 0.14$. The rough-wall boundary layer can be rescaled to model different

wind speeds [Castro (2007)].

The flow database consists of vertical planes of inflow to each rotor, which captures the wake aerodynamics generated by the upstream wind turbine(s). Therefore, the three velocity components are extracted in vertical planes of $2R \times 2R$ located one radius upstream of each turbine to reduce the turbine-specific influence of induction [Troldborg and Meyer Fosting (2017)], see Fig. 1. This corresponds to a grid of $39 \times 42$ points in the y-z plane. The time step is 0.1 seconds for simulations with

$U = 8, 12, 15 \ m/s$, and 0.05 seconds for $U = 20 \ m/s$. The data is extracted every 0.1 seconds during $2^{17}$ time-steps, which is approximately 3.64 hours of simulated flow.

**2.3 Parameter space and flow characteristics**

The database is designed to cover the majority of the operational range for this particular wind farm, and therefore the parameter space governing the turbulent wake flow. The most important parameter for wind turbine wakes is the thrust

coefficient $C_T$ [van der Laan et al. (2020)]:

$$C_T = \frac{T}{\frac{1}{2}\rho A U^2} , \qquad (1)$$

Where $T$ is the turbine's thrust, $\rho$ is the air's density, $A$ is the rotor's area, and $U$ is a representative velocity, typically the mean freestream axial velocity. This coefficient is a relative measure of the force exerted by the turbine with respect to the momentum of the incoming wind. For low wind speeds, the turbine extracts as much energy as possible, and the thrust coefficient is

typically around 0.8, which is considered high. Significantly higher values can result in flow reversal as the turbine enters propeller mode [Sørensen et al. (1998)]. For high wind speeds, the turbine typically pitches its blades to reduce power extraction and thrust force.

Four simulations were performed for different average incoming wind speeds, which cover a significant range of operating thrust coefficients. A second parameter inherently present in a wind farm is the turbine number (TN). As the flow enters the

wind farm, the incoming wind for the first turbine is undisturbed, but the second turbine operates in the wake of the first turbine. Further inside the wind farm, multiple wakes can be present concurrently. Wakes have a significant impact on the performance of wind farms, as the wind speed is lower and the turbulent intensity is higher causing a reduction in power production and increased fatigue loads on turbines operating in the wake [Vermeer et al. (2003); Porté-Agel et al. (2020)].

The parameter space covered by the database is visualized in Fig. 2. It consists of two parameters: turbine number (2-14) of

turbines operating in wake conditions, and four wind speeds at hub height for the front turbine (8, 12, 15, and 20 m/s).

Combined, these parameters are associated with a time-averaged $C_T$ of the upstream wind turbine, which generates the wake. $C_T$ is higher for low wind speeds until the turbine starts pitching. $C_T$ it is approximately constant through the wind farm at 0.8 and 0.3 for the cases of $U =8$ m/s and $U =20$ m/s, respectively. However, for $U = 12\text{-}15$m/s, there is a gradual transition in $C_T$ from the first turbine to the turbines further into the farm. Eventually, the flow will reach a balance between extracted power and wake recovery [Calaf et al. (2010)]. This is often referred to as the fully-developed or "infinite" wind farm, and is typically reached after the first 5-6 wind turbines [Andersen et al. (2020)]. Reaching the fully-developed wind farm flow essentially means that there is no discernible difference between the inflow to turbines operating deep inside the wind farm, and therefore a data point in the parameter space does not necessarily offer additional information. In total, Figure 2 shows 52 different combinations of the four wind speeds (U) and 13 turbine numbers (TN). Where each combination corresponds to a data set of inflow to a given turbine, $\mathbf{V}(y,z,t)$.

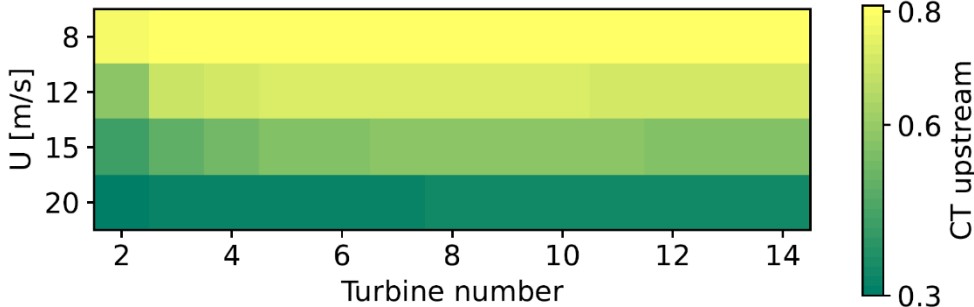

**Figure 2: Parameter space from the Large Eddy Simulations. $C_T$ is shown for the upstream turbine for each mean wind speed in the simulation inlet and turbine number.**

## 2.4 Proper Orthogonal Decomposition

Proper Orthogonal Decomposition (POD) is a classic technique for dynamic flow analysis, which decomposes a turbulent flow into modes of spatial variability. These modes are orthogonal and, given the norm used to perform the decomposition, optimal in terms of capturing the variance of the fluctuating flow [Lumley (1967); Berkooz et al. (1993)].

The velocity field ($\mathbf{V}$) is described as the sum of the mean flow ($\overline{\mathbf{V}}$) and the fluctuating flow ($\mathbf{V}'$), as in Equation 2.

$$\mathbf{V}(y,z,t) = \overline{\mathbf{V}}(y,z) + \mathbf{V}'(y,z,t) \,, \tag{2}$$

POD is then applied to the three fluctuating velocity components of $\mathbf{V}'$ ($u'$, $v'$ and $w'$), where each time step is represented as a column vector, and $N_t$ time steps are aggregated into a matrix $\mathbf{M} = [\mathbf{V}'_1, ..., \mathbf{V}'_{N_t}]$. The auto-covariance of $\mathbf{M}$ is computed: $\mathbf{R} = \mathbf{M}^T \mathbf{M}$, and the eigenvalue problem $\mathbf{RG} = \mathbf{G\Lambda}$ is solved, where $\mathbf{\Lambda}$ is a matrix of real and positive eigenvalues and $\mathbf{G}$ is a matrix of orthonormal eigenvectors $\mathbf{G} = [g_1, ..., g_{N_{t-1}}]$. The dimensionality has been reduced by 1 due to the extraction of the mean flow; and the orthonormality of the global modes is given using the standard inner product, $\langle \boldsymbol{a} ; \boldsymbol{b} \rangle = a_i b_i$, across all

flow components: $\langle \boldsymbol{g_i} ; \boldsymbol{g_j} \rangle = \delta_{ij}$. Finally, the modes are organized according to the eigenvalue decay i.e., in descending order according to variance, representing the turbulent kinetic energy contribution of each mode. Collectively, all modes form a new set of basis functions spanning the data set.

The original flow can be reconstructed by projecting the flow into each mode with a standard inner product, which results in its contribution as a function of time ($\phi_i$). Subsequently, as shown in equation 3, the modes multiplied by their contribution over time can be summed to reconstruct the flow.

$$\mathbf{V}'(y,z,t) \approx \sum_{i=0}^{K} g_i(y,z)\phi_i(t) \, , \tag{3}$$

An approximated reconstruction of the flow can be obtained by only including a limited number of modes ($K \leq N_{t-1}$).

## 2.5 Global POD basis

POD is traditionally applied on an individual flow case, i.e. on a "local" data set in the parameter space. Therefore, applying POD on a single data set is referred to as a *local POD basis* in the present work. The local basis contains the modes, which optimally represent the variance of that particular data set. Conversely, a *global POD basis* is formed by including multiple data sets in the decomposition [Andersen and Murcia Leon (2022)].

The global basis can be computed by including $q$ different datasets, and adding $N_T$ snapshots from each flow data set to the matrix $\mathbf{M}$ before applying POD:

$$\mathbf{M} = \left[\mathbf{V}'_{1,1}, \dots, \mathbf{V}'_{1,N_t}, \dots, \mathbf{V}'_{q,1}, \dots, \mathbf{V}'_{q,N_t}\right] \, , \tag{4}$$

Consequently, the global POD basis is sub-optimal at capturing the variance for a particular data set, but it is expected to provide a better representation across the entire parameter space.

## 2.6 Convergence of global POD basis

The expected sub-optimality of a global POD basis raises several questions on how effective a global basis is compared to a local basis . For example, how many datasets should be used and which datasets should be included to create a global basis with high-quality performance across the parameter space compared to a local basis.Here, the parameter space contains 52 datasets. This means that for any number of datasets $k$ composing a global base, there are $\binom{52}{k}$ possible global basis, so there are $\sum_{k=1}^{52} \binom{52}{k} = 4.5 \times 10^{15}$ possible combinations to generate a global basis, which effectively excludes the option of evaluating all of them. Consequently, the global POD bases are constructed in an iterative manner. First, a POD basis is based on a single dataset (one flow case in the parameter space), and its performance is evaluated across all flow cases of the parameter space. Secondly, a new flow case is added , and POD is applied to find the corresponding new basis, which is "global" because it was formed with more than one dataset. The new global basis is again evaluated across all flow cases before a new dataset can be added. In each iteration, the next dataset added to the decomposition corresponds to the flow case with

the maximum error across the parameter space, thereby maximizing the reduction of the overall error. The iterative procedure means that only 52 different combinations exist, as each data set can be chosen as the initial starting point.

## 3. Results

### 3.1 Flow cases

The wake flows change considerably across the parameter space. Figure 3 shows the normalized average streamwise velocity
and the turbulence intensity for the four corners of parameter space (Fig. 2).

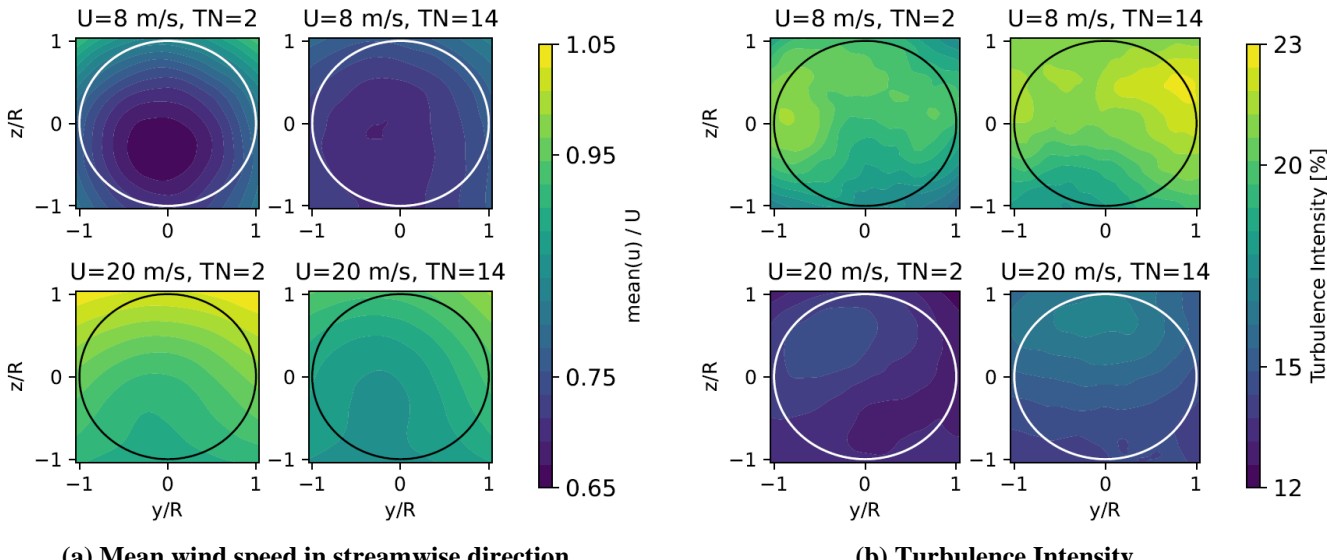

(a) Mean wind speed in streamwise direction.                 (b) Turbulence Intensity.

**Figure 3: Mean streamwise wind speed and turbulence intensity of the flows in the four corners of the parameter space. The circle on each plot represents the rotor.**


Figure 3a shows a significantly larger deficit and a more circular wake when $C_T$ is high (U=8 m/s), and a less significant wake and more dominant shear profile from the atmospheric boundary layer when $C_T$ is low (U=20~m/s). Furthermore, the spatial gradients are less pronounced late in the wind farm (TN=14), which is a consequence of the increased mixing due to the presence of multiple wakes. Figure 3b shows the streamwise turbulence intensity ($\sigma(u')/U$), which ranges from 12% up to
23% with the largest values in flows with a high thrust coefficient. The highest turbulence intensity is located in the upper half of the domain, where more momentum is exchanged between the wake and the surrounding atmospheric flow.

Figure 4 shows the streamwise velocity spectra taken at the rotor center for the four corners of the parameter space. This exemplifies how turbulent dynamics depend on both the thrust coefficient and turbine number. The total turbulent kinetic energy is larger for the high wind speed, as expected. The spectra tend to shift at the low frequencies, particularly for high $C_T$,
as the largest turbulent length scales are broken down as they move through the wind farm [Andersen et al. (2017)].

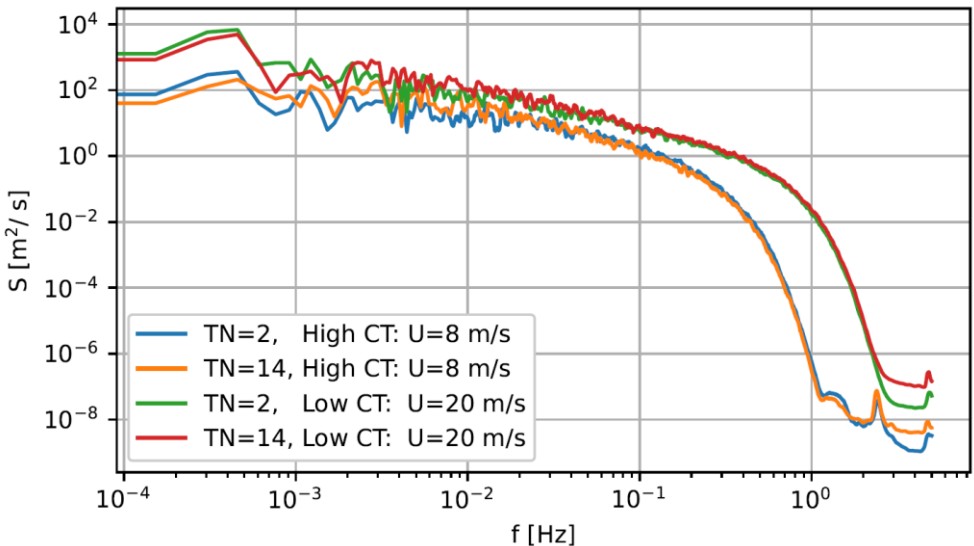

**Figure 4: Fourier spectra of u′ at hub height for the four corners of the parameter space.**

## 3.2 Global modes

POD is applied to compute the local and global POD bases. Figure 5 shows the first eight local modes calculated with one
dataset from the parameter space, 1P. The figure also shows eight global POD modes derived using 9 datasets, 9P. The local
and global modes are clearly similar, and are therefore capable of capturing the same coherent structures. However, the
ordering of individual modes might change as they cover an increasingly large parameter space. This is an important point of
the global basis. For instance, global mode 9P $g_7$ is not shown as it qualitatively corresponds to local mode 1P $g_9$, while global
modes 9P $g_7$ and 9P $g_8$ are more important over the parameter space. As shown by Andersen and Murcia Leon (2023), this
means that the contribution of variance captured by each mode might change over the parameter space.

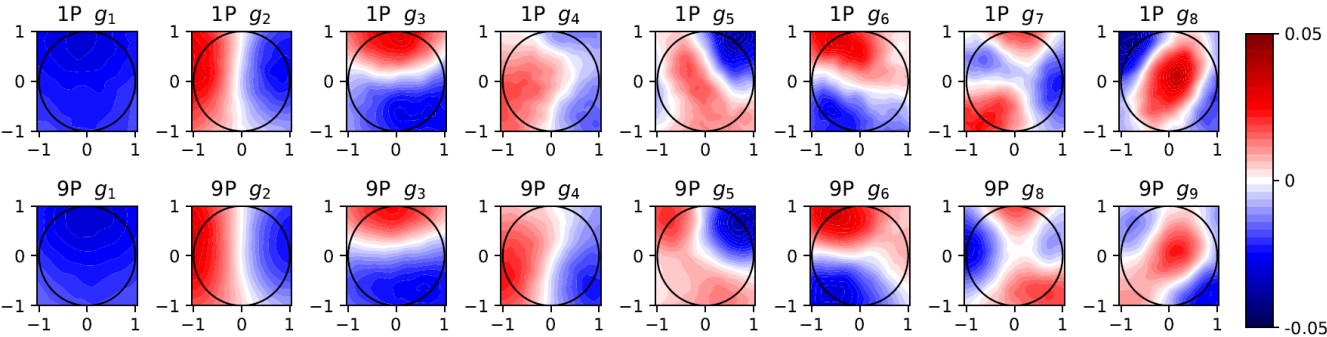

**Figure 5: Streamwise component of the first modes using one and nine points from parameter space, 1P and 9P respectively. The circle on each mode represents the turbine rotor.**

Although the local and global modes are qualitatively comparable, the global basis must be both efficient and representative of the entire parameter space. Figure 6 shows instantaneous flow fields for all velocity fluctuations $U'$, $V'$, and $W'$ for LES and reconstructions using the first 20 modes of P1 and P9 for flow case $U = 20 \, m/s$ and the 5th wind turbine, corresponding to the bases visualized in Figure 5. The filtering effect of POD is clearly seen in the reconstructions for both P1 and P9 for all velocity components, as the details of the LES are not reconstructed with only 20 modes. However, the overall structures of

the reconstructed flow fields are comparable, particularly for the streamwise fluctuations $U'$. The region of positive fluctuations (red) in $W'$ is slightly larger in P1, while P9 has a larger region without fluctuations (white) of $W'$. The figure also shows the difference in the instantaneous fluctuations from LES and the two reconstructions. The error fields of the two reconstructions are basically indistinguishable with only minor differences. Appendix A show the reconstructions and the corresponding errors using 8, 50, and 100 modes. The similarity in both reconstructed velocities and errors clearly shows that the two different bases

are equally efficient at reconstructing the flow for all practical purposes

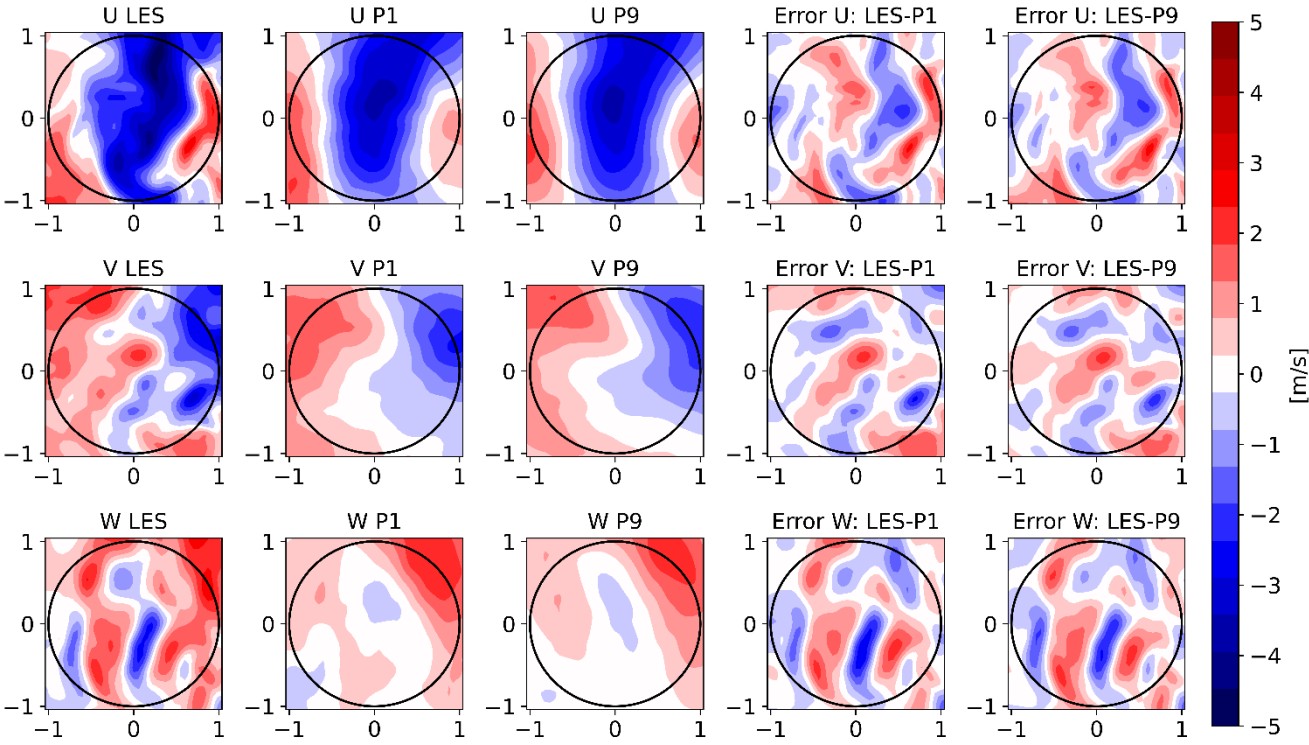

**Figure 6: Flow fields of LES, reconstruction using P1 and P9 as well as error computed as the difference between LES and the reconstructed flows using 20 modes for the 5th turbine and U = 20m/s. The top row shows streamwise velocity fluctuations $U'$, the center row shows lateral velocity fluctuations $V'$, and the bottom row shows vertical velocity fluctuations $W'$.**

### 3.3 Global modes convergence

In order to quantify the efficiency, a given basis is evaluated against the full LES flow using a velocity error $E_{vel}$ defined in equations 5 and 6. The metric takes, for every velocity component, the average in space (y,z) of the ratio between the root mean square error of the velocity field and its standard deviation over time. The error is normalized by the standard deviation because it is a direct measure of the variance in the original flow. Subsequently, $E_{vel}$ corresponds to the norm of the errors from the three velocity components. This results in a single value for each point in the parameter space that represents the total error of the reconstruction with respect to the original flow.

The velocity error is shown in Figure 7a, where the top left figure corresponds to the local basis using one dataset of U = 8m/s and TN = 14, indicated by the white number. This basis is evaluated across the entire parameter space using 100 modes, i.e. the basis derived from the one dataset is applied on all flows. It reveals that the velocity error is largest at the first turbines for U = 8m/s. However, the error of the reconstructed flow compared to the LES is low for the high wind speed. Hence, the global basis provides efficient reconstruction for significantly different flow cases.

$$E_{u\prime} = \text{mean}_{y,z} \left[ \frac{\sqrt{\text{mean}_t[u'_{LES}(y,z,t) - u'_{POD}(y,z,t)]^2}}{\text{std}_t[u'_{LES}(y,z,t)]} \right], \tag{5}$$

$$E_{vel} = \sqrt{E_{u\prime}^2 + E_{v\prime}^2 + E_{w\prime}^2}, \tag{6}$$

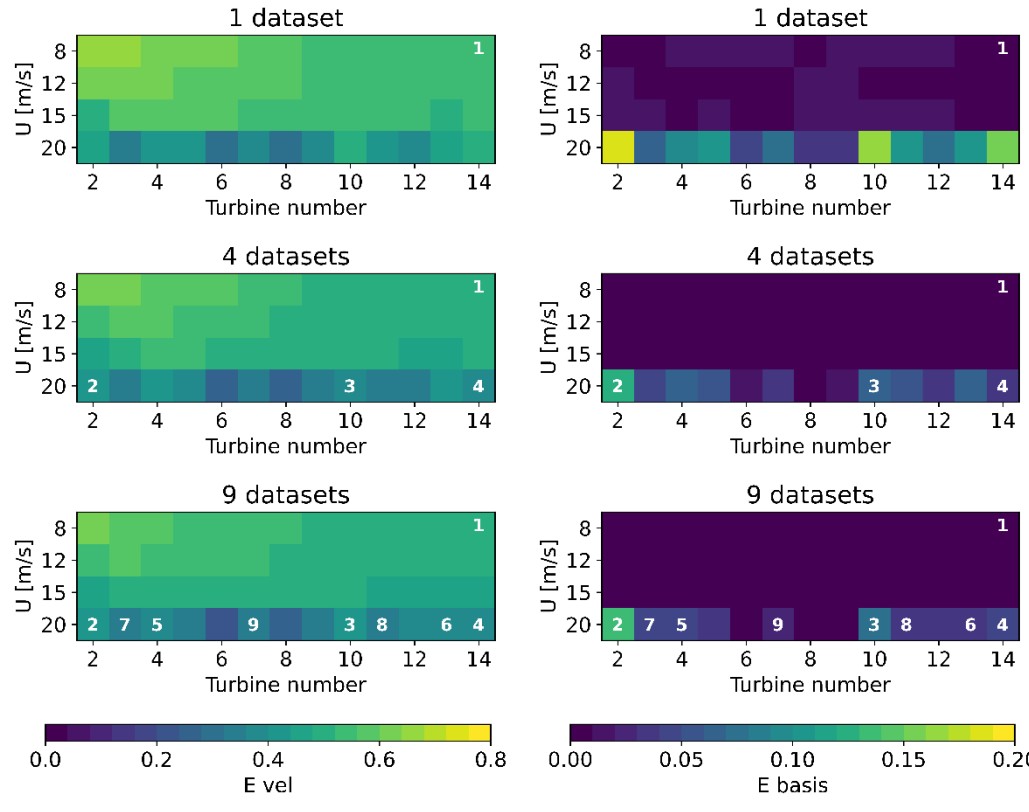

(a) **Velocity error fields in parameter space**       (b) **Basis error fields in parameter space**

Figure 7: Velocity error ($E_{vel}$) and basis error ($E_{basis}$) using 100 modes across the parameter space for basis including 1, 4, and 9 datasets.


The velocity error ($E_{vel}$) corresponds to the total error with respect to LES, but this has two components: a truncation error due to the number of modes included, and a basis error $E_{basis}$ due to the use of a global basis, which is non-optimal basis locally. The basis error arises because the global basis is sub-optimal compared to the local basis, which in principle is capable of reconstructing a larger portion of the flow with the same number of modes. Therefore, in order to isolate and quan4tify the

basis error, the velocity error from the local POD bases ($E_{vel-local\ POD}$) is subtracted from $E_{vel}$, as shown in equation 7.

$$E_{basis} = E_{vel} - E_{vel-local\ POD} ,\qquad(7)$$

The basis error is shown in Fig. 7b. Contrarily to the velocity error $E_{vel}$, the largest basis errors correspond to the low $C_T$. Hence, the dataset for U = 20 m/s and TN = 2 with the largest basis error is added to improve the global basis. The same error estimates are computed with the updated global basis, and the procedure is iterated to reduce the overall error of the global

basis.

Figure 7 shows the evolution of errors using four and nine datasets. The white numbers indicate the order of adding the different datasets to the global basis. The average errors are clearly reduced when more datasets are included. Additionally, the largest basis errors remain at low $C_T$, and the difference of including one or four datasets is significantly larger than using four or nine datasets, which suggests convergence of the global basis.

Figure 7 uses U=8 m/s and TN=14 as initial dataset for the iterative procedure. Table 1 shows five different starting points. The first four initial datasets correspond to the four corners of parameter space, and the fifth is a point in the middle of the domain.

| Coordinates | Name | | | | |
|---|---|---|---|---|---|
| | A | B | C | D | E |
| U [m/s] | 8 | 20 | 8 | 20 | 15 |
| TN | 2 | 2 | 14 | 14 | 5 |

Table 1: Initial datasets from parameter space.

Figure 8 shows the evolution of the average velocity error ($E_{vel}$) across parameter space as a function of the number of datasets in the global basis for the five different initial conditions. As seen, all five initial conditions (A-E) yield the same trend of decreasing the mean velocity error as more datasets are included. On average the mean velocity error decreases 6% from one to nine datasets. Effectively, the choice of the initial dataset increasingly loses importance as more datasets are included. For instance, with one dataset, the relative difference between the best and worst performing global basis is 3.9%, but with nine

datasets it is reduced to 0.4%. Furthermore, after including three datasets, the iterations starting at points B and D (those which started at U= 20 m/s) contain the same datasets, which means that from that point they yield the same results.

The examined bases A-E include 512 snapshots per dataset and the flow reconstruction was truncated at 100 modes. The horizontal lines in Fig. 8 indicate the average error when each flow in the parameter space is reconstructed with 100 modes of the corresponding local basis, computed using 512, 1024, and 2048 snapshots respectively. The performance of these bases

depends on the number of snapshots before achieving convergence, and the number of independent snapshots is limited, at around 2048, by the span of a single dataset. Independence is here based on snapshots being separated according to the integral length scales. On the other hand, a global basis can include more data because it is extracted from different datasets, i.e. different flows, which makes the snapshots independent.

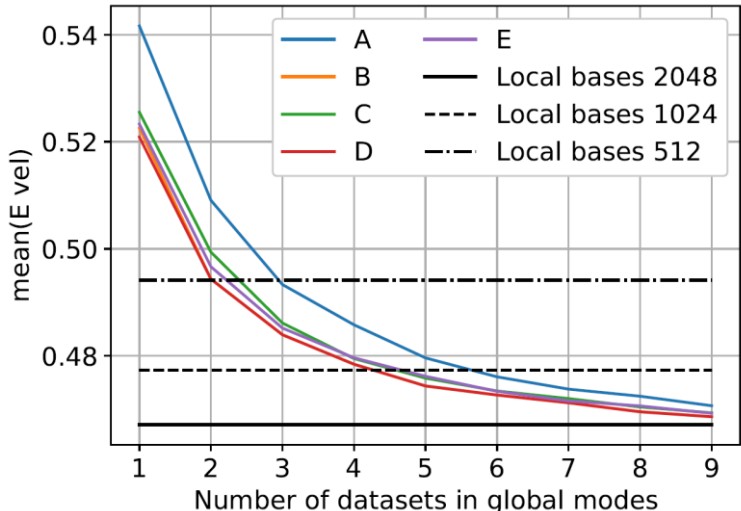

Figure 8: Mean error $E_{vel}$ across the parameter space using 100 modes vs the number of datasets included in the global basis.

Consequently, global bases can be directly compared to local bases when computed with the same amount of data. Table 2 compares the values of the horizontal lines in Fig. 8 (local bases error) with the average result from the curves A-E at the corresponding number of snapshots. As more datasets are included, the performance of the global bases gets closer to the theoretical minimum error of the local bases, where four datasets correspond to a relative difference in the error of 2.8% truncated at 100 modes.

| Total number of snapshots | Local bases error | Global bases | | Relative error difference |
|---|---|---|---|---|
| | | Number of datasets | Average error | |
| 512 | 0.494 | 1 | 0.527 | 6.7% |
| 1024 | 0.477 | 2 | 0.499 | 4.6% |
| 2048 | 0.467 | 4 | 0.480 | 2.8% |

Table 2: Mean velocity error comparison between local bases and global bases (average of curves A-E) with the same total number of snapshots using 100 modes.

For the dependence of the number of truncation modes, Fig. 9 shows the velocity error for two local and three global bases truncating at different numbers of modes. Overall, the error decreases as more modes are included. Here, it is also possible to compare the truncation error to the basis error. The basis error is approximately one order of magnitude smaller than the truncation error. For instance, using 100 modes, the error of the global basis using 1 dataset is 0.523, and the error of the best bases, i.e. local bases 2048, is 0.469, which is a relative difference of approximately 10%.

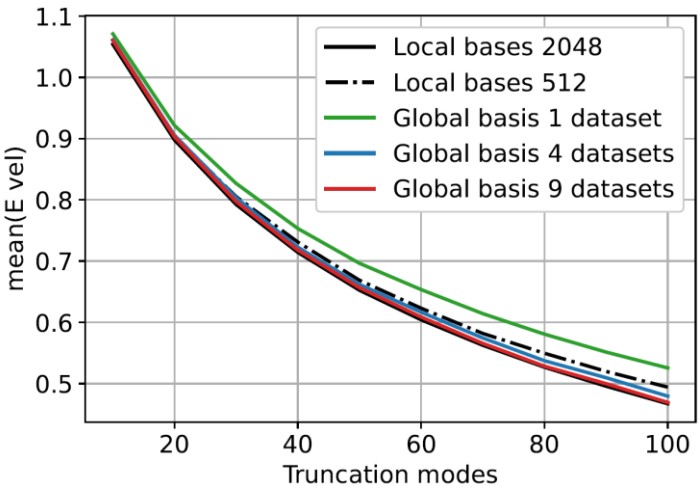

**Figure 9: Mean error E_vel across the parameter space vs number of truncation modes. The global bases shown correspond to the starting point C from table 1.**


It is noteworthy, especially for a global basis with a low number of datasets, that the basis error increases as more modes are included for the number of modes plotted. This is attributed to the fact that each additional mode from the optimal basis adds more information to the flow than a mode from the sub-optimal base. However, any of these bases are capable of completely reconstructing the flow if all modes are included. Therefore, it is expected that as more modes are included, the optimal bases

start to saturate, while the sub-optimal bases eventually will catch up and the basis error gap will reduce.

### 3.4 Case study with stochasticity

The physical and statistical implications of employing a global basis are investigated using the stochastic engine PS-ROM. The chosen global basis is generated based on four datasets shown in Fig. 7, which yield a basis error of 2.8%, see Table 2. The global basis is tested on the unseen flow case of $U = 12\ m/s$, *i.e.* the employed global basis does not contain information

from this flow scenario.

The inherent stochastic variability is assessed by generating $N = 30$ stochastic realizations and cross-comparing all realizations against themselves for a single flow case. This yields a total of $\binom{N}{2} = 435$ stochastic flow realizations.

A general spectral error metric $E_S$ of two spectrums is defined in equation 8:

$$E_{S-i,j} \equiv \frac{\int (\hat{S}_i - \hat{S}_j) df}{\int \hat{S}_j\, df}\,, \tag{8}$$

where $S_i$ and $S_j$ are the two spectrums to compare. $\hat{S}_i$ is $S_i$ filtered with a rolling mean using an averaging window varying logarithmically in size to smooth out higher frequencies.

The spectral error is utilized in two ways, where the analysis is shown for the streamwise velocity at hub height.

First, the variability of the 30 stochastic realizations is estimated to provide stochastic error distributions, as shown in Fig. 10 for each turbine. The red distributions are all centered around zero and show the stochastic variability of the 30 realizations

relative to themselves, *i.e.* how much can a single realization of a constructed flow scenario vary relative to numerous realizations of the same flow. The distributions tend to narrow further into the wind farm, which indicates how the deep farm flows become increasingly self-organized and governed by the wakes [Andersen et al. (2017)].

Second, the development can also be examined by comparing the spectral error between different flows, i.e. comparing 30 stochastic flow realizations at each turbine against the inflow to a specific turbine. Here, the last turbine (TN=14) is chosen to

represent the fully-developed wind farm. The spectral error given by equation 8 is also used for this comparison, which yields $N^2 = 900$ error samples since both flow cases have N=30 stochastic realizations. The corresponding errors are shown as green distributions in Fig. 10. The distributions for the first turbines are significantly offset with a negative error, but initially narrower than the stochastic distributions in red. Eventually, the green distributions gradually become centered around zero. Hence, the distributions of the stochastic spectral error and the spectral error relative to the 14th turbine can be compared

directly to determine if there is a statistical difference between inflow to a given turbine relative to the last turbine. If the error distributions are reflections of each other, it implies that there is no statistical difference between the velocity spectra at the center of the domain between the turbine number in question and turbine number 14. This is particularly useful when trying to determine if the flow dynamics have reached the fully-developed wind farm conditions, where the statistical distributions no longer change as the turbine number increases [Andersen et al. (2015)]. This is the case from approximately TN=9 forwards.

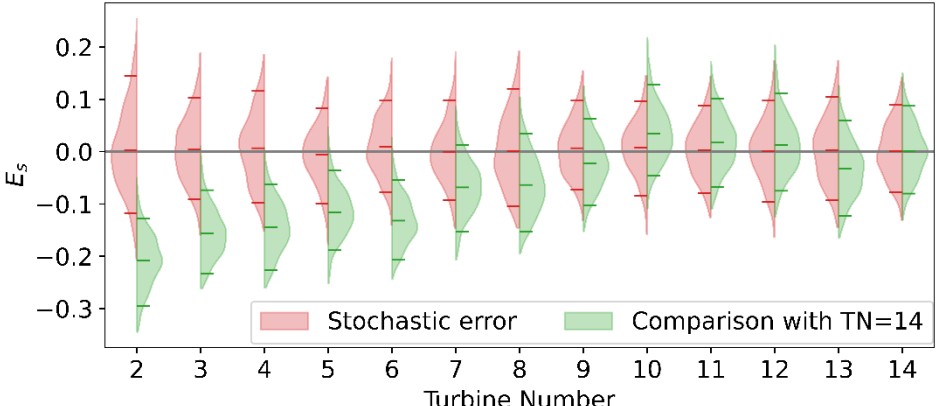


**Figure 10: Distributions of spectral errors for the streamwise velocity at the rotor's center related to the stochasticity and relative to turbine number 14 through the wind farm using 30 stochastic realizations and 100 global POD modes for the simulation case U=12 m/s.**

The contribution of individual modes to the different flow cases can also be compared. Figure 11 shows similar spectral error distributions for global modes 2, 5, 12, and 20. Modes 2 and 5 show a gradual evolution reminiscing of Fig. 10, where the

distributions gradually become increasingly similar. Contrarily, higher modes, such as 12 and 20, present a more scattered behaviour with lower errors and mean values varying between positive and negative for TN > 2 indicating more stochastic behaviour. This suggests that the transition to a deep wind farm state is primarily dictated by changes in the first global modes,
which are associated with the largest turbulent scales. This trend corroborates the findings of Andersen and Murcia Leon (2023), where it was clearly shown how different global modes are active in different locations within the wind farm to capture different flow scenarios, e.g. turbines operating in freestream conditions, turbines operating in single wake, or turbines operating in fully-developed conditions.

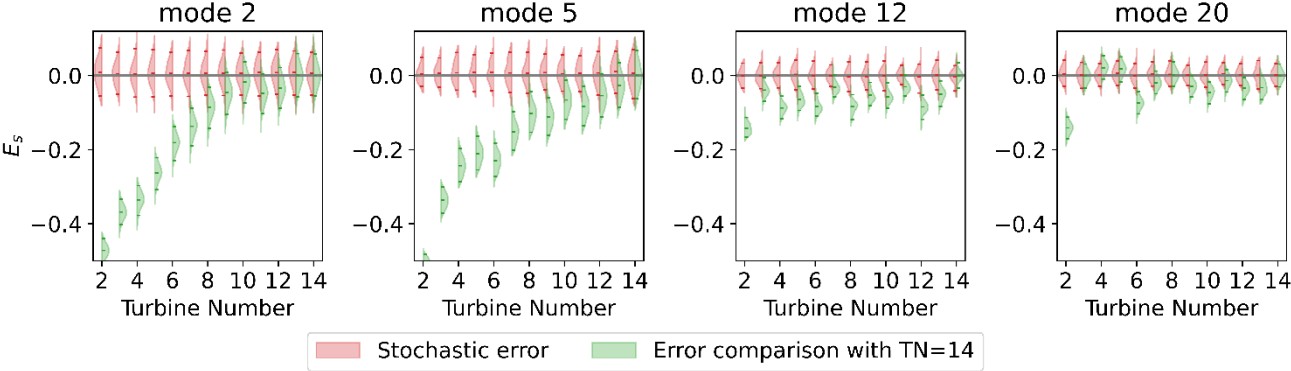

**Figure 11: Distributions of spectral errors for four modes related to the stochasticity and relative to turbine number 14 through the wind farm using 30 stochastic realizations for the simulation case U= 12m/s.**

## 4 Discussion

Employing a global POD basis allows the use of the same modes for an entire parameter space but introduces the basis error
in the flow reconstructions (equation 7). The basis error emerges because the global basis is not as efficient at reconstructing a particular flow as the local POD basis. However, it was shown that this error is reduced as more datasets are included in the global basis, and it is approximately one order of magnitude smaller than the truncation error for 100 modes.

The convergence of the global POD basis as a function of the number of datasets has a parallel with the convergence of the local POD basis. A local POD basis converges when enough snapshots of the flow are included, so it contains information
about all the dynamics that occur in the flow [Hekmati et al. (2011)]. Consequently, it is an optimal basis in terms of variance, and including more snapshots would not improve its performance. Similarly, for the global POD basis, as more datasets are added, more information on the dynamics of the flows across parameter space is included. On the limit, when sufficient information covering the parameter space has been supplied, including additional datasets with flow dynamics that have already been covered will not improve the performance of the basis.

The inclusion of more datasets in the global basis implied adding more data in total, as the number of snapshots per dataset was kept constant. An alternative would be to keep the amount of snapshots constant, and hence include fewer snapshots per dataset; which would reduce the time to compute the modes that scales quadratically with the number of snapshots. This is an unexplored scenario, but it is speculated that it would only be useful as long as there are enough snapshots per dataset to generate an acceptable local POD basis with them, which would imply that there is enough information per dataset to capture

its dynamics.

The systematic process of including addition datasets is focused on minimizing the basis error (equation 7) with respect to the local bases' performance. The iterative procedure particularly identifies that more datasets from flows corresponding to low $C_T$- should be included. However, additional datasets could be identified using alternative metric, e.g. the velocity error, which would prioritize adding datasets of low turbine numbers and high $C_T$. Applying such alternative metrics, or simply selecting

datasets arbitrarily in the parameter space, also results in a reduction of the error as the number of datasets increases and therefore the global basis will eventually converge on multiple error metrics. However, it might be impractical to perform a detailed and systematic convergence study of the global basis for all applications. Yet, the present analysis shows how global bases are relatively insensitive to which datasets are used. It is therefore generally recommended to select multiple datasets, which represent various key flow phenomena. Selection of datasets apriori would typically require domain knowledge to

identify key scenarios with different physics, e.g. single wake, multiple wakes as well as different $C_T$-values.

Furthermore, the case study highlights a number of benefits of employing a global basis. The global basis enables detailed and quantifiable physical interpretation of how the flow changes within the parameter space, as also seen in Andersen and Murcia Leon (2023), where the modal statistics of a global POD basis applied on a full wind farm clearly reveal three main flow regimes of atmospheric inflow, single wake and multiple wakes. The expansion of the parameter space reveals new insights

compared to Andersen and Murcia Leon (2022). For instance, it is clearly seen how the spectral error distributions converge further into the wind farm indicating when fully-developed wind farm flow dynamics are achieved and how this is linked to the first few modes (Figure 11). The method also enables modelers to estimate both the impact and uncertainty of different flow realizations as well as different modes when generating synthetic turbulent flows. Additionally, the analysis reveals how wind turbine wakes are relatively coherent flows, which can be covered by approximately 100 modes. Although, the

consistently larger basis error for U= 20m/s also highlights how more modes are required to reduce the errors for undisturbed atmospheric flows, where the influence of the turbines is negligible.

Finally, the global bases are used to model and analyze highly turbulent wind turbine wakes and the present work expands the parameter space to cover two dimensions compared to the single parameter in Andersen and Murcia Leon (2022). In principle, there are no limits to the number of dimensions. However, it is speculated that the efficiency of global bases will significantly

decrease if the parameter space covers multiple dimensions with very different flow cases. If so, more modes would be required for the flow generation. However, the efficiency and convergence of the linear global POD bases also gives promises that it is possible to utilize nonlinear dimensional reduction techniques, such as autoencoders, to increase efficiency further, *i.e.* reduce

the number of modes required [Brunton and Kutz (2019), Lee and Carlberg (2019)]. Therefore, global bases are expected to be generally applicable for dimensional reduction within fluid dynamics.


## 5. Conclusions

Wind turbine wake aerodynamics are inherently complex and chaotic, thus making accurate modeling and analysis of their dynamics particularly challenging. One approach is to decompose the flow using POD, which gives an orthogonal basis of spatial modes. The spatial modes can provide physical insights to the largest coherent structures and the modes can be used to developed reduced order models. The modes are optimal in terms of capturing the variance, and the original flow can be reconstructed as the sum of a truncated set of modes, which fluctuate over time. However, different flows can result in different modes, which makes it difficult to construct general reduced order models as well as compare different flows to provide insights to the physical differences. These caveats can be overcome by utilizing a global basis, where multiple flow cases are combined.

Global POD bases are shown to efficiently capture wind turbine wake aerodynamics for a parameter space covering all wake-affected turbines in a large w during different operating conditions (thrust coefficients). The performance of the global basis has a basis error with respect to the optimal local POD basis. However, the error is one order of magnitude smaller than the truncation error, which can be remedied by including a few additional modes. Most importantly, the basis error is significantly reduced and the efficiency convergence towards the local POD basis as more datasets are included to construct a global basis.

The efficiency is shown to be rather insensitive to the selection of flow cases to include in the construction of the global basis, especially when the basis error is compared to the truncation error of the flow reconstruction or the stochastic variability of flow realizations. However, it is recommended to include key features from the different flows in the parameter space.

Global bases also provide a consistent baseline for direct comparison of different flow cases and thereby enable physical interpretability of the flow behavior across the parameter space. For example, the evolution of the modes through the wind farm reveals that only the first few modes are responsible for the transition to a deep wind farm state, while higher modes corresponding to smaller turbulent structures mainly provide stochastic variations.

The convergence and benefits of global bases are illustrated here in the context of analyzing wind turbine wake flows with reduced-order models. However, the results are expected to generally apply to other turbulent flow scenarios, physical interpretation and model development are challenging.

## 6. Acknowledgements

Computational resources have been provided by the DTU cluster Sophia. DTU Computing Center Technical University of Denmark (2022) and the work has been partly funded by DTU Wind Energy through the Wind Farm Flow CCA 2022 as well

as the MERIDIONAL project (https://meridional.eu/) with grant agreement no. 101084216 under the European Union's Horizon 2020 research and innovation programme.

## 7. Code availability

The codes EllipSys3D and Flex5 are available with a license.

## 8. Data availability

Subsets of the datasets can be made available by contacting the authors.

## 9. Competing interests

The authors declare that they have no known competing financial interests or personal relationships that could have appeared to influence the work reported in this paper.

## 10. Author contribution

The three authors conceived the idea of this article based on previous work developed by Andersen and Murcia Leon. The LES simulations were done by Andersen, and Cespedes did the POD calculations and convergence of the global modes based on a code previously developed mainly by Murcia Leon. Finally, the writing and revision of this article was equally distributed among the three authors.

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

## Appendix A: Flow Reconstruction and Errors

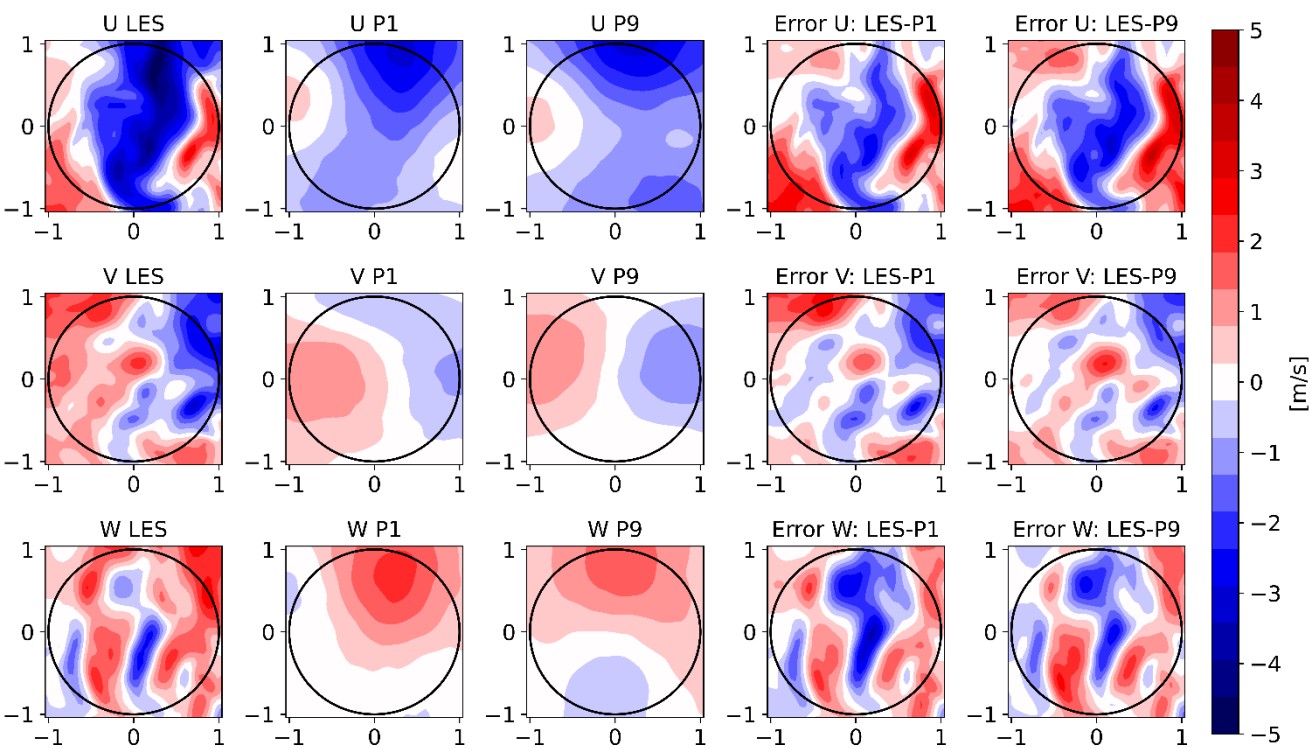

**Figure 12 A: Figure 6 using 8 modes**

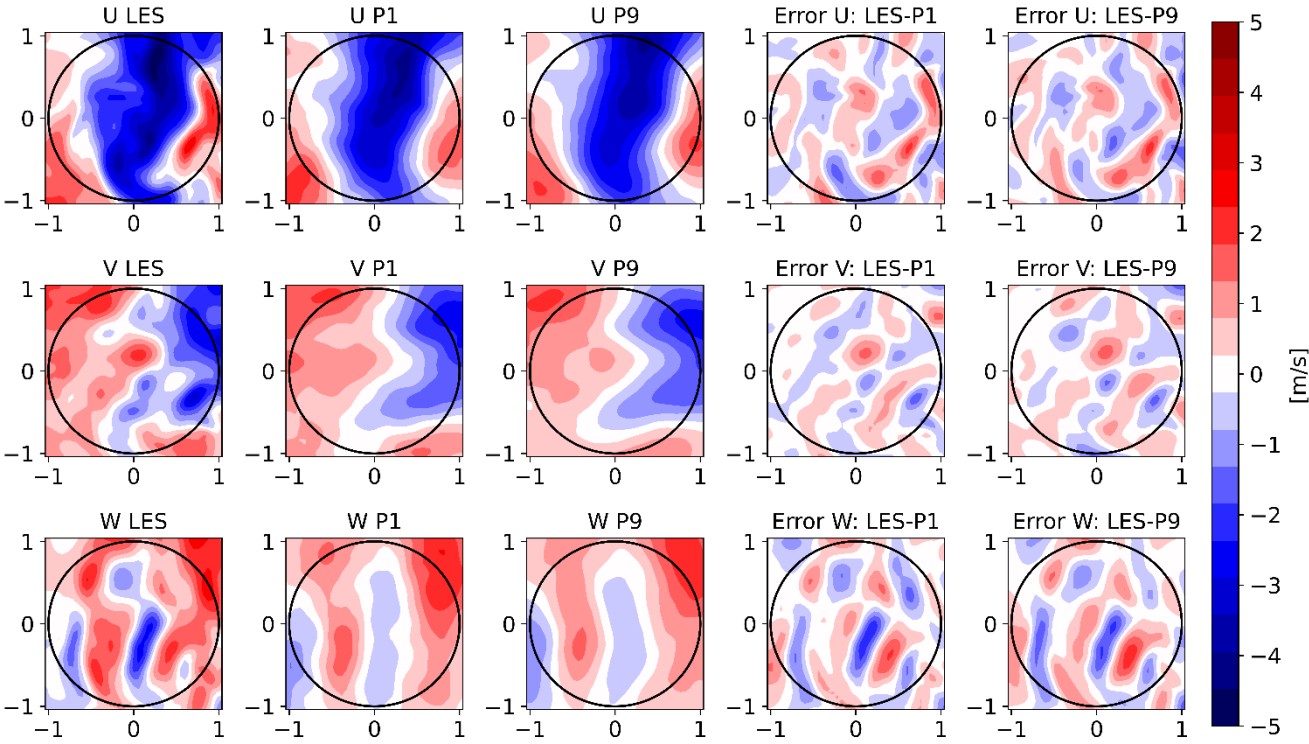

580

**Figure 13 A: Figure 6 using 50 modes**

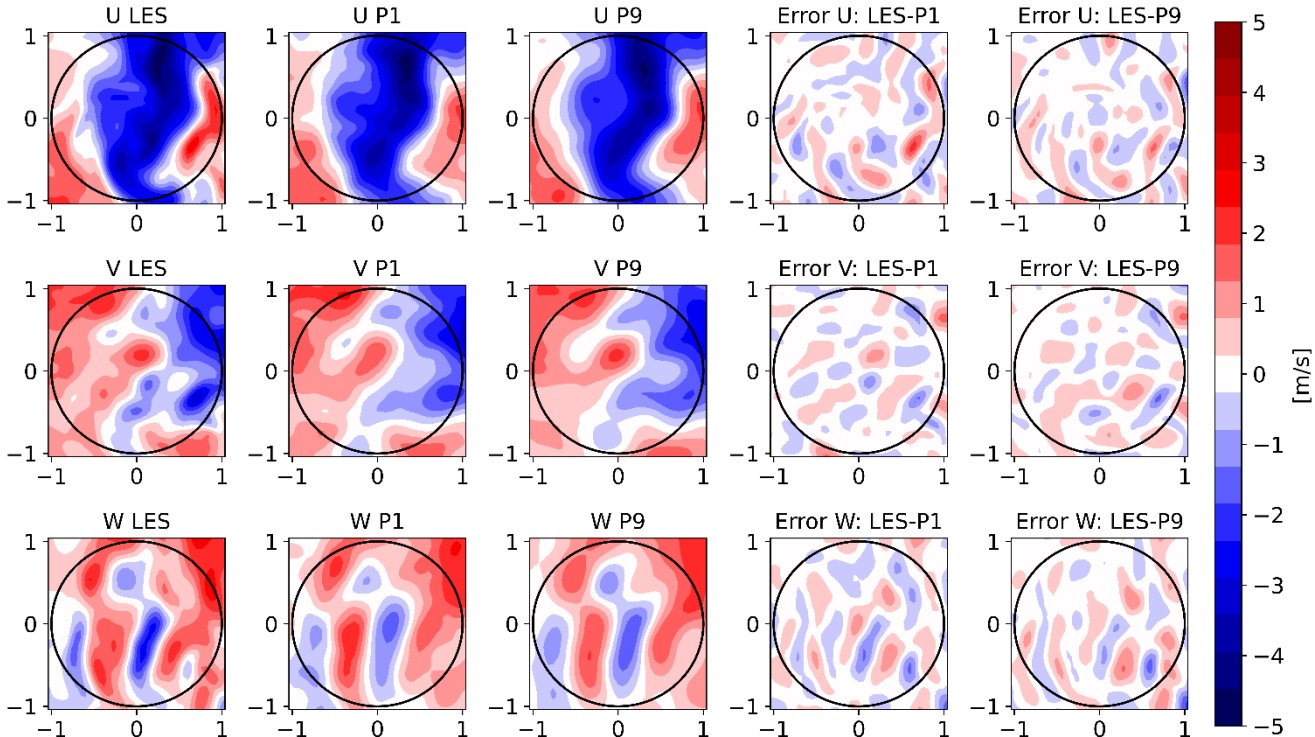

**Figure 14 A: Figure 6 using 100 modes**

585

