# Peer review of "Convergence and efficiency of global bases using Proper Orthogonal Decomposition for capturing wind turbine wake aerodynamics"

_Wind Energy Science, 2024_

## Author Response (AR1)

**Response to Referee #1: Convergence and efficiency of global bases using proper orthogonal decomposition for capturing wind turbine wake aerodynamics**

**Overall Comments**

*Dear Authors,*

*I have reviewed your article and would like to provide my feedback on it.*

*The article investigates the performance of the Proper Orthogonal Decomposition (POD) method when applied to a dataset that is obtained combining data from different flow cases. A dataset of the flow inside a wind farm is used as the case study.*

*I think the topic of the article can be interesting for scientists working on fluid dynamics. However, there is a lack of significant developments that are specific to wind energy research, which is the topic of this journal. The objectives, most of the methodology and a large portion of the results are about the POD method itself and not about wind energy systems. So, I think this article is not entirely appropriate for this journal and, in its current state, is more suited to a journal on fluid dynamics.*

*I would also say that article is not very well written: many sentences, particularly in the Methodology section, are hard to follow and it's difficult to understand their content.*

*In the Discussion section, it is hard to understand what information is important. In the Conclusion there isn't any message specific to the wind energy field.*

*In reason of these comments, I am against publication of this article. My suggestion is to revise the text to improve clarity of presentation. Then, if Authors would like to discuss the POD in general, I suggest considering a journal of fluid dynamics, otherwise they should make it clear which is the impact and usefulness of this research for the wind energy community.*

XXXXXXOnce again, thank you for the comments. This document contains written responses to each of the comments with links to changes in the paper, and we include our intermediate response previously posted online here in red. We have had particular focus on improving the Discussion and Conclusion.

Dear Reviewer,

Thanks for the comments. The specific comments mainly center around rephrasing for clarity, which we will address later. For now, we will just briefly address the overall comment of the review to provide additional background on the content and overall motivation of the article.

We fully acknowledge that the article is somewhere in between "applied wind energy" and "general fluid dynamics methodology". In fact, we have previously submitted the article to a more "fluid dynamic" journal, and we also believe that it has applications beyond wind energy as outlined in the Discussion. However, the article was rejected as it was viewed as too wind energy-specific. Hence, we have now resubmitted it to WES.

As the references show, other researchers have utilized global bases, but global bases have to our knowledge not been applied within wind energy before the development of PS-ROM (Andersen and Murcia Leon, 2022). Any newly developed model should in our opinion be continuously tested in

order to expand or verify the application, which is particularly important for data-driven models. This article extends the previous application beyond changes in CT to also encompass all turbines on a single basis, i.e. freestream, single waked and deep wake flows, which is a significant step toward generalizing the model. Wake aerodynamics are characterized by highly turbulent flows, and therefore we present new spectral metrics to quantify the efficiency across a range of conditions, particularly focusing on the optimality typically seen as a major advantage of POD although we here show it is less important than truncation. Finally, presenting a method to quantify the error related to stochastic realizations also has general implications for wind energy. Overall, a fast dynamic wake model delivering LES accuracy is in our view a significant contribution to the wind energy community.

Best regards,

Juan Felipe, Juan Pablo and Søren

**Specific Comments**

*As said in the General comments, there are many sentences that are not clear and should be revised. These sentences are:*

We have rephrased and clarified all sentences.

*- First sentence of the abstract. This sentence, in the position it has (the first one of the article) is obscure. I suggest adding a few lines to introduce the article content.*

We have rephrased the abstract and added more content to the first sentences.

*- Line 35-36. These two sentences are not clear. You must explain what a single flow case is and the difference of considering multiple flow cases.*

It was clarified what it means a single flow: *"the resulting bases are typically applied to data from a single flow case, which corresponds to a single point in a parameter space. A single flow case could for example be the flow around a cylinder at a specific Reynolds number or in the present case the inflow to a particular wind turbine in a wind farm operating at a single $C_T$ value."*

*- Line 46, "alongside . . . parameter space". Not clear. This is the introduction. Try to give enough information so that any reader can understand.*

We have rephrased the paragraph for clarity.

*- Line 81, "which captures the wake dynamics". Maybe you mean the wake of the upstream turbine? This is not clear, because a turbine inside the line is affected by wakes of all upstream turbines.*

"Wake dynamics" reference to the wake generated by the turbines upstream. The text has been clarifed.

*- Line 101-103, "Ct of the wake-generating . . . in the wake".*

The description of the parameter space has been clarified.

*- Line 104, "the operation of . . . of the wake"*

The text has been clarified.

*- Line 106-107, "where the operating . . . wake recovery".*

Changed to *"where the flow has achieved a balance between extracted power and wake recovery"*

*- Line 109-110. Isn't Fig.2 showing the Ct? Explain the relation between Ct and V.*

We have clarified the explanation of the parameter space.

*- Line 149-150, "Consequently, ... in the parameter space".*

The sentences has been rephrased for clarity.

*- Line 179-180, "these are not ... POD modes".*

We have rephrased the sentence.

*- Line 201-202, "which indicates ... different flows". Not clear, must be argued better.*

A low error means that the flow is reconstructed efficiently despite the basis being derived from another dataset. We have rephrased for clarity.

*- Line 211, "The basis error ... of modes".*

Clarified.

*- Line 232-234, "It is noted ... 0.4%".*

Clarified.

*- Line 82: "one radius upstream". The effect of induction is present also at a higher distance from the rotor. Maybe you should rephrase this sentence to explain what you mean with "reduce the influence of induction" and why you did not take the plane further upstream.*

Yes, induction extends further upstream than $1R$, what we meant to say is that the induction will be approximately turbine agnostic further upstream. We have clarified with reference to Troldborg and Meyer Forsting, 2017.

*- The Data availability section is missing. Code and data should be made available to the public to reproduce the results of the article.*

We have added sections on code and data availability.

**Technical Corrections**

*Line 68-69, "The grid ... outlet boundaries". If it is important, maybe it's worth adding a second panel in Fig. 1 with a sheme of the grid.*

Thanks for the suggestion. We have elaborated on how the equidistant region is, but as it is a cartesian grid with simple stretching, we do not think a figure is not very informative nor necessary.

*Line 90: "and U is the velocity", what velocity?*

We have elaborated on this velocity.

*Line 97: replace "atmospheric" with "undisturbed".*

Fixed.

*Line 274: of PS-ROM presented by Andersen and Murcia Leon (2022)*

Fixed.

*Line 299: you should use "" to make it (Andersen et al. (2015))*

Fixed.

**Response to Referee #2: Convergence and efficiency of global bases using proper orthogonal decomposition for capturing wind turbine wake aerodynamics**

**Overall Comments**

*The present manuscript tackles a potentially interesting topic for wind energy such as the development of a general basis for model reduction using POD. The subject is interesting and relevant for wind turbine's research, but there are some points that need to be clarified or developed.*

Thank you for the comments to the paper. We have clarified aspects and provided responses to your comments below in blue.

*First of all, the paper does not discuss in detail the contest of model reduction for wind turbines. In particular, the introduction lacks reference to the many previous works on POD analysis of wind turbine and wind farm flows (among others, VerHulst and Meneveau Physics of Fluids 2014, Bastine et al. Energies 2015, Hamilton et al., Wind Energy 2015 Physics of Fluids 2016, Phys. Rev. Fluids 2017, Wind Energy 2018; De Cillis et al. Wind Energy 2021, Renewable Energy 2022, Journal of Physics: Conference Series 2022). Also, DMD and ANN are mentioned but not put in the contest of wind turbine flows (see, for instance, Debnath et al. "Towards reduced order modelling for predicting the dynamics of coherent vorticity structures within wind turbine wakes" 2017; De Cillis et al. "Dynamic-mode-decomposition of the wake of the NREL-5MW wind turbine impinged by a laminar inflow" 2022, among many others).*

We have added some references using POD for wind turbine wakes, but we have limited the references on DMD and ANN to general articles/reviews without providing a complete review as these methods are not used in the present paper, but merely mentioned as alternative approaches.

*Moreover, I have some concerns regarding the convergence of the grid, which seems to me too coarse for the LES of 14 turbines, having only 20 points per radius in each direction. The grid convergence of the LES for the considered case needs to be shown in the manuscript, maybe in a dedicated appendix.*

20 cells per radius is very highly resolved for simulations of wind farms with actuator discs. A full grid study is beyond the scope of this paper, but we have provided reference to Hodgson et al., 2023, who compared two state-of-the-art numerical solvers, including EllipSys3D used here. Hodgson et al. concluded grid requirements to be larger than 6.5 cells per radius, so we have 3x times the required minimum. Furthermore, typical design using the IEC-61400-1 standard corresponds to turbulent boxes with resolution of 32x32 covering the rotor, while these simulations has a resolution of 40x40 covering the rotor.

*The same can be said about the choice of the time step set for the extraction of the snapshot from the LES, as well as the total time of the simulation. This quantities are known to affect considerably the convergence of the POD algorithm over one single dataset, so the convergence with respect to this parameters need to be discussed for some of the considered dataset.*

The simulation time steps are chosen in accordance with standard requirements for actuator disc simulations to not conflict with CFL-conditions, where we maintain CFL less than 0.5. In principle, POD should be based on uncorrelated snapshots, i.e. separated by 2 integral length scales or more, but if the snapshots are not uncorrelated they will not add new information and eigenvalues will be 0 (machine precision). As stated the total simulation is 3.64 hours, which is very long for this type of simulation, and the simulations are statistically stationary. The insights into the convergence required by the reviewers are provided in Figure 7, Table 2, and lines 236-250 in the original manuscript, where the number of snapshots used for the local basis is varied and the number of independent snapshots is estimated to be 2048. We have clarified what independent snapshots mean. The alternative to changing the number of snapshots in each dataset would be to compare the convergence for a fixed number of snapshots, which was outlined the original discussion.

*Moreover, I cannot see any information about the value of the tip speed ratio, and about the whether tower and nacelle are taken into account in the simulation, which are both very relevant information for the considered flow case . In case tower and nacelle are not taken into account, I suggest to discuss the relevance of the simulations to realistic conditions, and to reformulate using the work "rotor" instead of "turbine".*

The actuator discs are fully coupled to Flex5, which gives dynamic estimations of loads and deflections. This also means that the turbines have a dynamic controller and therefore adjust the tip-speed-ratio relative to the local inflow according to the controller. We have added that nacelle and tower are not modelled, but we disagree with the reviewer that this exclusion should significantly impact the realism of the simulations with reference to the work by Zahle and Sørensen, who conclude that it has an influence of less than 2% based on blade resolved simulations. The influence on the resulting wakes would only decrease further into the row of wind turbines as the turbulence intensity increases.

*Also, the discussion about the choice of the global POD basis is not sufficiently developed. I understand that the global basis is constructed in an iterative manner, but I am not sure what does this means exactly. For instance, the performance of a basis is evaluated only using the velocity error? Which is the condition for adding a dataset? And why the dataset with worst performance should be added? Since the construction of the global basis is a crucial point of the paper, it needs to be discussed in much more detail. The same can be said about the case study with stochasticity, which is not sufficiently clearly explained. For instance, the authors should explain in more detail the fact that "the actual projected spectrums are used.." etc.*

We have clarified the iterative process for adding new datasets, where new datasets to be added are selected as the dateset with the largest error. It is an obvious choice as the global basis is intended to be efficient over the entire database and by adding a dataset with the highest error, the overall error is reduced the most and the global performance is improved. However, the Discussion also includes comments that the iterative process could be based on adding datasets based on alternative metrics, such as the velocity error, or arbitrary datasets.
The flows are projected into the global basis in the case study with stochasticity, and these spectrums are used. The alternative would be to predict the spectrums of unseen cases as done in Andersen and Murcia Leon, 2022. We have restructured the paragraph and clarified the text.

*Finally, I think that the performance of the POD basis cannot be measured only using an integral quantity such as the velocity error, since even if the integral error is rather low, the flow field might have some important structural differences with the simulated one. The POD-reconstructed flow fields need to be shown and compared with the LES snapshots at given times, by showing a velocity error field for each velocity component.*

Thanks for the good suggestion. We have added Figure 6, which show a very good reconstruction

of all three velocity components.

**Typos**

*page 2, line 59: Large Eddie –¿ Large Eddy*

Fixed.

*page 2, line 49: the acronym LES need to be defined*

Fixed

*page 13, line 275: spectrums –¿ spectra*

Fixed

---

## Author Response (AR2)

**Message to editor on: Convergence and efficiency of global bases using proper orthogonal decomposition for capturing wind turbine wake aerodynamics**

Dear Cristina,

Please find attached our two responses to the second review of our article. As you can tell, we have decided to push back on some of the comments by the second reviewer, as we think they are misplaced and would make the article unfocused in our opinion. We have provided elaborate justifications on these points and why we do not include them as requested in the article.

Best regards,

Juan Felipe, Juan Pablo and Søren

**Second response to Referee #1: Convergence and efficiency of global bases using proper orthogonal decomposition for capturing wind turbine wake aerodynamics**

*Dear Authors,*
*After a second review of your article, I believe it has improved since the initial revision. The Abstract and Introduction now outline the article's relevance to the wind energy community. The text is clearer and somewhat easier to read.*

Thank you for the comments to the paper. We have clarified aspects and provided responses to your comments below in blue.

*The article can be still improved in some points (lines numbers refer to the ATC document):*

*- Abstract: briefly explain what is a global POD base.*

We have clarified the definition of a global POD basis, i.e. that it is based on multiple flow cases.

*- Line 137: "and it is approximately constant (trough the wind turbines in the farm) at 0.8 and 0.3".*

Fixed. "through the wind turbines in the farm" was added.

*- 144: "The fully-developed or "infinite" wind farm is typically reached after the first 5-6 wind turbines [Andersen et al. (2020)]." This concept is applied to discuss the findings in section 3.4. I suggest clarifying the idea of fully-developed wind farm flow and its relation to assessing the performance of the POD reconstruction technique.*

Fully-developed conditions are beneficial if there is no discernible difference between inflow to for example turbine number 8 and turbine 9. Hence, even a local POD basis would be similar for both turbines, and therefore obviously a global basis will quickly converge to cover these conditions.

*- 181: "The expected sub-optimality of a global POD basis raises a number of central questions on the effectiveness relative to a local POD basis and on the required number and which datasets to include in the construction of a global POD basis". This sentence is not clear (the required number of what?). Please clarify it.*

Re-phrased to: "The expected sub-optimality of a global POD basis raises several questions on how effective a global basis is compared to a local basis. For example, how many datasets should be used and which datasets should be included to create a global basis with high-quality performance across the parameter space compared to a local basis"

*- 187: "The new global basis is again evaluated". Isn't it "the POD resulting from the new global basins is again evaluated"?*

No, the POD is not resulting from the new global basis. Instead, POD is used to create the new global basis (global because it includes more than one dataset). Then this new basis is evaluated

again across the parameter space. This has been clarified:

'[...]. Secondly, a new flow case is added, and POD is applied to find the corresponding new basis, which is "global" because it was formed with more than one dataset. The new global basis is again evaluated across all flow cases before a new dataset can be added.'

- *189-190: "the dataset is added to the decomposition is the one with the worst performance to maximize the reduction of the overall error". Please clarify this sentence.*

The text has been clarified: "the next dataset added to the decomposition corresponds to the flow case with the maximum error across the parameter space. The reduction of the overall error is maximized with each iteration by including data from the specific flow case with the largest error."

- *Figure 6: make the horizontal and vertical axes equal so the rotor becomes a circle.*

Implemented.

- *234-239: "is difficult to qualitatively assess that P1 is significantly better than P9." Recall the objective of your evaluation and why it matters that P1 does not surpass P9.*

The key point is not that P1 can not surpass P9. In mathematical terms, P1 will be slightly better than P9, as P1 is optimal at capturing the variance. The point is that the two bases are equally good for all practical purposes. We have rephrased to: "The error fields of the two reconstructions are basically indistinguishable with only minor differences. The similarity in both reconstructed velocities and errors clearly shows that the two different bases are equally efficient at reconstructing the flow for all practical purposes."

- *378: "how different global modes are actuated". I think the verb "actuated" is wrong here.*

We have change the verb to "active", but the intended meaning is the same. The point is that different flow scenarios are captured by different modes, where some modes are very important for capturing certain flow conditions. This "activation" of different modes is shown in Figure 11, and we refer to similar analysis in Andersen and Murcia, 2023.

- *402: "The iterative procedure particularly identifies adding more datasets with low CT". Clarify this sentence.*

Changed to: "The iterative procedure particularly identifies that more datasets from flows corresponding to low $C_T$ should be included"

- *411: "Selection of datasets apriori would typically require domain knowledge to identify key scenarios with different physics", thus it can be impractical (?).*

Requiring domain knowledge is in our opinion not impractical, but simply implies that selecting datasets apriori would require physical understanding of the flows, e.g. knowing that freestream, single wake and multiple wake corresponds to different physics.

- *423: "indicating when fully-developed wind farm flow conditions are achieved dynamically and how this is linked to the first few modes". Clarify.*

We have elaborated previously how the fully-developed wind farm flow conditions relate to the convergence of the global POD basis. This convergence is clearly shown in the convergence of the distributions of selected modes (Figures 11). Once, converged it is a clear indication that the dynamics of the flows are comparable.

- *425: different (flow) realizations*

Implemented.

*- 433: "However, the efficiency and convergence of the linear global POD bases also gives promises that it is possible to utilize nonlinear dimensional reduction techniques, such as autoencoders, to increase efficiency further, i.e. reduce the number of modes required". Explain better what you mean.*

We have included a couple of references on how non-linear methods such as autoencoders can provide more efficient dimensional reduction compared to POD.

*- 439: "parameter space". Recalls which are the parameters.*

Changed to: "[...] for a parameter space covering all wake-affected turbines in the wind farm during different operating conditions (thrust coefficient)."

*- 443: "The performance of the global basis has a basis error with respect to the local POD basis". Rewrite this sentence.*

Changed to: "The global basis gives an error compared to the optimal local basis. However, [...]"

*- Conclusions need to be revised. Begin by summarizing the problem you aim to address, with straightforward explanations about wind farm physics. Next, highlight the tools employed in the study and summarize their key performance findings. Finally, discuss anticipated future advancements in research on this topic.*

The conclusion was re-structured keeping this comment in mind.

**Second response to Referee #2: Convergence and efficiency of global bases using proper orthogonal decomposition for capturing wind turbine wake aerodynamics**

*The manuscript's clarity has been improved. However, I have still a few comments about the modified parts, that should be taken into account.*

Thank you for the comments to the paper. We have clarified aspects and provided responses to your comments below in blue.

*- abstract: " Wind turbine wake aerodynamics ,,, but are highly turbulent". I think this sentence should be rephrased (The aerodynamics are turbulent?)*

Rephrased for clarity.

*- page 3: "nacelle or tower, but this only has a minor influence.." I do not agree with this statement, since De Cillis et al. Wind Energy (2021) have proven that the presence of the tower have a strong influence on the POD modes in the wind turbine's wakes. They used actuator line modeling instead of actuator disk, but since the former model is more accurate then the latter, their conclusion is still to be taken into account and mentioned. Moreover, the nacelle has been found to have a strong effect in the generation of low-frequency oscillations of the wake, such as wake meandering, by many other studies (see for instance, the recent experimental work of Biswas & Buxton, J. of Fluid Mechanics, 2024, among many other previous studies). I thus ask for a deeper discussion on this important point.*

In our opinion, the reviewer overestimates the importance of towers on the flow in large wind farms, particularly in terms of importance for this article. First, after the first review, we included a reference to Zahle and Sørensen, 2008, who performed blade-resolved simulations of a single wind turbine, including a tower, in uniform inflow and sheared inflow. Blade resolved simulations are higher fidelity than both actuator discs and lines as used by De Cillis et al., and despite the lack of inflow turbulence Zahle and Sørensen conclude that the tower has limited influence $(1-2\%)$, which we deem to be "minor influence". With increasing fidelity also comes increasing computational costs, and it is not feasible to perform LES large wind farm simulation with blade resolved turbines nor is it academic standard to include towers in LES of large wind farms. Second, the results in De Cillis et al., albeit interesting, are also based on simulations without inflow turbulence and a relatively low tip-speed ratio of 3. The tip-speed ratio and inflow turbulence have a significantly higher impact on the wake development, see, e.g., Equation (4.12) in [1]. The main impact of the tower is to introduce asymmetry (as also noted by Biswas and Buxton), which initializes the tip vortex instability and breakdown. Once the tip vortices have broken down, the wake transitions to small-scale turbulence and eventually far wake[1], where distinct tip vortices and tower wake will have disappeared. This is also clearly seen in the experimental results by Biswas and Buxton, see for instance Figure 4, where the Gaussian fit is very good for most distances, at least from $x/D > 1.5$, a clear indication that the wake has transitioned. In comparison, the present simulations are for turbine spacing of $6D$. Biswas and Buxton comment on how practical

limitations result in experimental towers and nacelles with a relative size of 3 times larger than real-life turbines. Clearly, experimental campaigns can still give important insights to the physics, but the impact of oversized tower and nacelle will be overestimated (Biswas and Buxton, page 980). Despite this overestimation, Biswas and Buxton also clearly state that tip vortices dominate in the near-wake, not the tower wake. In highly turbulent conditions such as wakes, the breakdown outlined above will happen even faster and reduce the impact of tip vortices and towers. Third, the method, analysis and metrics are generic and does not depend on these numerical details, but could as well be applied on a dataset including towers and nacelle. To summarize, if the reviewer had provided specific references on the importance of modeling tower and nacelle in the highly turbulent conditions inside wind farms, then we would happily include it. But the suggested references do not sufficiently substantiate the importance to warrant a deeper discussion in the context of our article, which focus on the development and efficiency of reduced order models and how global POD can be used to analyze and compare various flow conditions, where we already refer to the interesting paper by De Cillis et al. in the context of using POD to provide physical interpretability.

*- Figure 6: "all velocity components as the details of the LES are not reconstructed with only eight modes". I agree with that, but what is the point of showing a bad reconstruction that does not structurally reflects the real flow? The figure clearly shows that the error is of the same order of the velocity (unless I miss some normalization in the last two columns, it seems that the error reaches around 100% of the LES velocity value), This is indeed a proof that a global measure is not a sufficiently good way to ensure the level of convergence of the reconstructed flow. The authors should demonstrate that their POD basis is able to provide a fair reconstruction, built up using more modes, able to reproduce also the finer vortical structures of LES, otherwise the whole comparison among the used POD basis, is pointless.*

First, let us remind the reviewer of their comment in their first round:
*Finally, I think that the performance of the POD basis cannot be measured only using an integral quantity such as the velocity error, since even if the integral error is rather low, the flow field might have some important structural differences with the simulated one. The POD-reconstructed flow fields need to be shown and compared with the LES snapshots at given times, by showing a velocity error field for each velocity component.*
We acknowledged this good idea and have therefore included Figure 6, which adds significant value to the manuscript. However, we fail to understand why the reviewer now thinks this is 'pointless' compared to the first review. To us, the point is that the figure clearly shows that the two reconstructions as well as the two errors are basically indistinguishable, i.e. the global POD basis provides a reconstruction, which is as good as the local POD basis. Therefore, the magnitude of the error is not important as we can always include more modes to reduce the error (see Figure 9). We have intentionally chosen to only include a few modes to emphasize how well the global basis performs, which in our opinion is a direct and fair comparison based on the initial suggestion by the reviewer. Furthermore, we note that the turbine itself acts as a spatial filter [2], so the finer vortical structures are less important for a reduced order model aimed at providing wind turbine load calculations. However, the higher modes will tend to capture more isotrophic scales, which both the local and global basis will capture. So even though, smaller vortices are less important for our application, we include the same figure for 8, 20, 50, and 100 modes in appendix and below for the reviewer. As seen, the performance of 1P and 9P is the same for all reconstructions, while the error decreases as more modes are included.

*- Figure 7 and related discussions: as stated in the last comment, a global measure is not a sufficiently good way to ensure the level of convergence of the reconstructed flow. Thus, the authors should evaluate the convergence of the POD basis also using local measures, such as for instance the percentage of the maximum or rms error $dU$ with respect to $U_{mean}$.*

We believe we have substantiated this sufficiently in our previous answer, that the global POD basis provide spatial reconstructions, which are indistinguishable from the local POD basis, and therefore a global measure is appropriate. The article already discusses that different metrics could be used to evaluate and identify which dataset to add. As described in the article, our motivation for the chosen metric is that 'it is a direct measure of the variance in the original flow', where local POD is optimal in terms of capturing the variance. Again, the overall aim is to show that global POD bases are approximately as efficient as local POD bases, but come with additional advantages in terms of constructing reduced order models and providing physical insights to different flows.

**References**

[1] Jens Nørkær Sørensen, Robert Mikkelsen, Dan S. Henningson, Stefan Ivanell, Sasan Sarmast, and Søren Juhl Andersen. Simulation of wind turbine wakes using the actuator line technique. *Royal Society of London. Philosophical Transactions A. Mathematical, Physical and Engineering Sciences*, 373(2035):20140071–20140071, 2015.

[2] S. J. Andersen and J. P. Murcia Leon. Predictive and stochastic reduced-order modeling of wind turbine wake dynamics. *Wind Energy Science*, 7(5):2117–2133, 2022.

[Figure]

Figure 1: Figure 6 from the article, flow fields of LES and reconstruction using P1 and P9, using 8 modes.

[Figure]

Figure 2: Figure 6 from the article, flow fields of LES and reconstruction using P1 and P9, using 20 modes.

[Figure]

Figure 3: Figure 6 from the article, flow fields of LES and reconstruction using P1 and P9, using 50 modes.

[Figure]

Figure 4: Figure 6 from the article, flow fields of LES and reconstruction using P1 and P9, using 100 modes.

---

## Author Response (AR3)

**Message to editor on:**
**Convergence and efficiency of global bases using proper orthogonal decomposition for capturing wind turbine wake aerodynamics**

January 16, 2025,

Dear Cristina,

We hope this message finds you well.
The current submission has been revised in accordance with the requested changes. Specifically, minor typos have been corrected, and the reference format has been updated as per the guidelines

Best regards,
Juan Felipe, Juan Pablo and Søren